# Raman Spectroscopy Profiling of Splenic T-Cells in Sepsis and Endotoxemia in Mice

**DOI:** 10.3390/ijms241512027

**Published:** 2023-07-27

**Authors:** Ibukun Elizabeth Osadare, Ling Xiong, Ignacio Rubio, Ute Neugebauer, Adrian T. Press, Anuradha Ramoji, Juergen Popp

**Affiliations:** 1Institute of Physical Chemistry (IPC), Abbe Center of Photonics (ACP), Friedrich Schiller University Jena, Member of the Leibniz Centre for Photonics in Infection Research (LPI), Helmholtzweg 4, 07743 Jena, Germany; ibukun.osadare@uni-jena.de (I.E.O.); ute.neugebauer@med.uni-jena.de (U.N.); juergen.popp@ipht-jena.de (J.P.); 2Department of Anesthesiology and Intensive Care, Jena University Hospital, Friedrich Schiller University Jena, Am Klinikum 1, 07747 Jena, Germany; ling.xiong@med.uni-jena.de (L.X.); ignacio.rubio@med.uni-jena.de (I.R.); adrian.press@med.uni-jena.de (A.T.P.); 3Center for Sepsis Control and Care (CSCC), Jena University Hospital, Friedrich Schiller University Jena, Am Klinikum 1, 07747 Jena, Germany; 4Leibniz Center for Photonics in Infection Research, Jena University Hospital, Friedrich Schiller University Jena, 07747 Jena, Germany; 5Leibniz Institute of Photonic Technology, Member of Leibniz Health Technologies, Member of the Leibniz Centre for Photonics in Infection Research (LPI), Albert-Einstein-Straße 9, 07745 Jena, Germany; 6Faculty of Medicine, Friedrich Schiller University Jena, Kastanienstraße 1, 07747 Jena, Germany

**Keywords:** sepsis, Raman spectroscopy, infection, inflammation

## Abstract

Sepsis is a life-threatening condition that results from an overwhelming and disproportionate host response to an infection. Currently, the quality and extent of the immune response are evaluated based on clinical symptoms and the concentration of inflammatory biomarkers released or expressed by the immune cells. However, the host response toward sepsis is heterogeneous, and the roles of the individual immune cell types have not been fully conceptualized. During sepsis, the spleen plays a vital role in pathogen clearance, such as bacteria by an antibody response, macrophage bactericidal capacity, and bacterial endotoxin detoxification. This study uses Raman spectroscopy to understand the splenic T-lymphocyte compartment profile changes during bona fide bacterial sepsis versus hyperinflammatory endotoxemia. The Raman spectral analysis showed marked changes in splenocytes of mice subjected to septic peritonitis principally in the DNA region, with minor changes in the amino acids and lipoprotein areas, indicating significant transcriptomic activity during sepsis. Furthermore, splenocytes from mice exposed to endotoxic shock by injection of a high dose of lipopolysaccharide showed significant changes in the protein and lipid profiles, albeit with interindividual variations in inflammation severity. In summary, this study provided experimental evidence for the applicability and informative value of Raman spectroscopy for profiling the immune response in a complex, systemic infection scenario. Importantly, changes within the acute phase of inflammation onset (24 h) were reliably detected, lending support to the concept of early treatment and severity control by extracorporeal Raman profiling of immunocyte signatures.

## 1. Introduction

Sterile inflammation and infection might present similar cardinal signs in humans, yet inflammatory and infectious diseases can exhibit different degrees of severity [1]. One of the key players is the host’s immune system, which reacts to inflammation and infections in various ways, depending on the underlying cause, and focuses on restoring normal functions and maintaining homeostasis [2]. Different methods are employed for detecting infection or inflammation by investigating circulating cellular biomarkers in the blood or phenotyping the immune cells. Various biochemical and surface modifications provide information on the immune cell status [3]. For example, the analysis of immune cells on a single-cell level has been extensively used in recent years for developing targeted and tailormade diagnostics and therapeutic options for personalized medicine. An example of this is the recent importance of single-cell RNA sequencing to harness heterogeneity within immune cell subsets of septic mice [4] and in humans to enable the study of differentially expressed genes between healthy and pathological conditions or for automated cell-type annotation to gain insights into immune dysregulations during sepsis [5].

Similarly, the single-cell transcriptomic analysis identified hyperinflammatory cell subtypes where the transcriptomic signature of monocytes varies among viral sepsis patients [6,7]. However, the overall functional changes and their implicating biochemical status are inadequately understood. The T-lymphocytes especially play a decisive role in coordinating the immune response toward recognizing various pathogens and are also involved in many inflammatory diseases [8,9,10]. Despite intensive research in the last decades, much remains to be learned about the T-cell response in severe human infections. Understanding the immunological responses of T-lymphocytes during endotoxemia and septic insult can be vital in developing a targeted drug therapy. This can further address the current unmet clinical need for precision medicine requiring a rapid diagnostic tool for detecting the presence and type of infection. Rapid screening and differentiation between inflammation and infection have high therapeutic significance, as the latter requires an anti-infective intervention. Hence, the aforementioned cellular information can be gained by phenotyping the cells in their native state with the minimum possible invasive interventions and on a single-cell level preferably label-free.

Raman spectroscopy is currently emerging as a novel biophotonic tool that delivers biochemical information intrinsic to a cell on a single-cell level [11,12]. It is label-free, nondestructive, and enables determining the functional and chemical states of cells [13]. In Raman spectroscopy, the interaction of light with the biomolecules results in inelastic scattering of the light. The inelastically scattered light provides information on molecular vibrations intrinsic to the molecule. Hence, the different peaks observed in the Raman spectra provide complex information about the molecular composition of the cell. Raman spectroscopy, therefore, allows capturing information on the molecular composition. This method has been used in characterizing leukocytes for single-cell analysis [14,15], to follow the cellular activation status in cell lines [14,16,17], and cell–drug and cell–cell interactions [11,16,18], as well as cell differentiation and polarization [11,19,20]. In the last decade, Raman spectroscopy has been extensively applied to investigate the immune cell activation status [11,20,21,22,23,24,25,26,27,28]. Previously, Raman spectroscopy was applied to follow time-dependent biochemical changes from acute to post-acute inflammation stages in the T-lymphocytes of endotoxemic mice. The aim was to gain insight into the potential of Raman spectroscopy for distinguishing biochemical alterations in endotoxemia. The study showed the possibility of using Raman spectroscopy to follow the T-cell activation process post-LPS (lipopolysaccharide) treatment in a rodent model of endotoxemia [11,22]. The Raman spectroscopy results indicated that T-lymphocytes regain their primary biochemical composition, similar to the control cells, after LPS shock for 30 days [22,29]. Endotoxemia results in general inflammation in the body due to exposure to LPS, a component of Gram-negative bacterial cell walls [22,29]. In the case of polymicrobial infection-induced sepsis in mice, T-lymphocytes cause systemic inflammation, with their functionality severely hampered and organ dysfunction leading to sepsis [30,31]. In the current work, the chemical phenotype of splenic T-lymphocytes from mice subjected to a peritoneal infections sepsis model (PCI) is compared with that seen in endotoxemic shock. The peak of infection and significant changes in the splenocytes are expected to be visible [27]. Knowing specific immune responses in individual splenic cells during sepsis and endotoxemia will be advantageous for understanding the disease progression and for identifying therapeutic targets.

## 2. Results and Discussion

The T-lymphocyte population isolated from the spleens of endotoxemic and septic mice was investigated to gain insight into the potential of Raman spectroscopy for distinguishing biochemical alterations in endotoxemia or sepsis. Previous studies showed that T-lymphocytes regain their functionality after an acute endotoxemic episode [22,29], whereas T-lymphocyte functionality is severely hampered during sepsis [32]. Endotoxemia results in general inflammation in the body due to LPS exposure, a component of Gram-negative bacterial cell walls [22,29]. In the case of polymicrobial infection-induced sepsis in mice, the T-lymphocyte causes systemic inflammation and organ dysfunction leading to sepsis [30]. In the current work utilizing Raman spectroscopy, the aim is to capture the distinct response of T-lymphocytes manifesting as a result of the differences in intracellular biochemical profiles. The averaged Raman spectral profile of T-lymphocytes from the septic mice and endotoxemic mice, along with the healthy controls (sham mice), are generated for overall comparison. In the mean Raman spectra, prominent Raman spectral features are observed in both the sham and infected animals, i.e., CH stretching region at ~2936 cm^−1^, amide I vibrations in the range 1649–1658 cm^−1^, and the nucleic acids vibrations around 785 cm^−1^ and 1094 cm^−1^. In the mean Raman spectra, subtle differences can be seen. However, it is hard to comprehend the differences present between the sham and infected mice.

### 2.1. Comparison of Raman Difference Spectrum between Mice with Sepsis/Inflammation and Sham

The Raman difference spectrum shown in Figure 1 highlights the differences between the T-lymphocytes from the control and sick mice. The mean Raman spectra of sepsis mice (Figure 1A) and endotoxemic mice (Figure 1B), along with their respective control groups, are shown in Figure 1. The Raman difference spectra are generated by subtracting the mean Raman spectra of the control group from the sick group, the Raman difference spectrum for the septic mice is shown in Figure 1C, and the Raman difference spectrum for the mice with endotoxemia is shown in Figure 1D.

The spectral difference observed for the septic mice is one order of magnitude higher than the endotoxemia mice, as can be visualized by the *y*-axis scale displaying the Raman peak intensity. In the difference spectrum of the septic mice, the contributions of Raman peaks in sick mice are from nucleic acid bands (1094 cm^−1^ and 1334 cm^−1^), proteins (1295 cm^−1^), lipids (2873 cm^−1^, 2891 cm^−1^), and CH stretching protein vibrations (2942 cm^−1^ and 2963 cm^−1^). In comparison, the healthy cells exhibit amino acids (527 cm^−1^ and 641 cm^−1^); proteins (620 cm^−1^, 1634 cm^−1^, and 1676 cm^−1^); lipids (1154 cm^−1^, 1538 cm^−1^, 1754 cm^−1^, and 3038 cm^−1^); and nucleic acid vibrations at 1517 cm^−1^ [33].

In the Raman difference spectrum for LPS, notable differences are visible in the regions representing amide I and the CH stretching vibrations. It can be observed that there are minimum changes in the intensity of the Raman bands between the sham and LPS-treated mice compared to the differences observed between the sham and septic mice. In the inflamed mice T-lymphocytes, no notable nucleic acid bands were registered compared to the healthy mice. This difference can be attributed to the different immune responses and progression of the biochemical changes that occur in the infected cells based on the mode of infection, type, and virulence of the infecting organism [21,34].

To visualize the differences between the sepsis and inflamed mice, the Raman difference spectrum was generated, where the mean Raman spectra of the mice with inflammation (LPS treatment) were subtracted from the septic mice (PCI), as shown in Appendix A. The difference spectrum highlighted prominent peaks indicating biochemical differences occurring within the inflamed and septic cells.

Some of the significant contributions can be seen in the regions of amino acids (647 cm^−1^, 743 cm^−1^, and 839 cm^−1^); nucleic acid vibrations (776 cm^−1^, 1175 cm^−1^, and 1574 cm^−1^); and amide I vibrations (1655 cm^−1^) for the septic mice (PCI-infected mice). On the other hand, the T-lymphocytes from the endotoxemic mice have a contribution from the amino acids (815 cm^−1^), nucleic acids (893 cm^−1^), and proteins (1013 cm^−1^ and 1349 cm^−1^) in the fingerprint region and a substantial contribution coming from the CH stretching region assigned to the lipids and proteins (2900 cm^−1^, 2951 cm^−1^, and 2961 cm^−1^). The significant difference in the Raman spectra of T-lymphocytes belonging to the PCI- and LPS-treated mice relates to the nucleic acids and protein vibrations. The chemical interpretation of the observed Raman bands is summarized in Table 1. The T-lymphocytes from endotoxemic mice exhibit rich CH stretching regions compared to the septic mice features (Appendix A), although the difference between the healthy mice from PCI and the endotoxemia model also exhibits high-intensity CH stretching vibrations, albeit with less pronounced spectral features (Appendix A). It is important to note that the variations between the control mice of the PCI and endotoxemia models extend beyond differences in the mice strains. The difference in the CH stretching region among the healthy mice of the PCI and endotoxemia models is due to the heterogeneity within individual healthy mice demonstrated via the Raman difference spectrum generated for experimental batches 3 and 4 of the PCI model (Appendix A), as well as the difference spectrum for endotoxemic control mice in experimental batches 2 and 3 (Appendix A). Notably, these observed differences remain consistent for all control mice within the PCI and endotoxemic models, as depicted in Appendix A. In septic mice, the T-lymphocytes are enriched with nucleic acid vibrations. The enrichment in the nucleic acid Raman signal in the T-lymphocytes of the septic mice indicates inflammation. In our previous study, Raman imaging of the T-lymphocytes enabled us to visualize morpho-chemical images of the nucleus. The activated T-lymphocytes exhibited a disintegrated nucleus compared to the control cells, resulting in abundant Raman signals arising due to the DNA/RNA vibrations [22].

Furthermore, the recent literature reports that mitochondrial DNA influences the immune system, causing an inflammatory response during sepsis [32,35,36]. These observations have been evidenced mainly in monocytes/macrophages. However, it is known that T-lymphocytes respond to mitochondria to trigger adaptive immune responses [37], and upon its activation, there is an increase in the mitochondrial mass and corresponding mitochondrial DNA [38]. Therefore, the elevated transcriptomic activity in septic T-lymphocytes is the probable cause for the higher nucleic acid signals observed in the Raman spectra of septic mice.

**Table 1 ijms-24-12027-t001:** Raman spectral band assignments [39,40,41].

Tentative Assignments of the Raman Peaks (cm^−1^)				
Nucleic Acids	Proteins	Lipids	Lipids and Proteins	Amino Acids
755 764 776 779 782 788 893 982 1085–1106 1175–1196 1304–1334 1517–1589	620 764 1004–1043 1127 1229–1295 1337–1349 1409–1463 1604–1676 2942–2984	1064 1130–1166 1367–1376 1463 1532–1538 1742–1766 2849 2873–2900 3011 3023	2918 2915	527 554 596 641 647 659 695 743 815–839

### 2.2. Comparison of Raman Models Built for T-Lymphocytes from Endotoxemic and Septic Mice

The multivariate analysis determined the significance of the Raman spectral differences between the sham and endotoxemic/septic mice. The multivariate analysis combined unsupervised PCA and different supervised multivariate analyses—namely, the LDA, RF, and SVM methods—to differentiate between sham and infected mice (for both the endotoxemic and PCI infection models). It was possible to differentiate the septic mice from the sham mice using all three statistical methods. The balanced accuracy obtained after 10-fold cross-validation is displayed in Appendix A, providing an overview of the performances of the statistical methods for the mice in each experimental batch. We chose the widely applied PCA-LDA statistical model for differentiating T-lymphocytes from the sham and sick mice to build a combined model including all the mice from different experimental batches. For the PCI mice, to enable internal (leave-one-batch-out) and external (using an independent test data set) cross-validation of the PCA-LDA model, batches 2, 3, 4, and 5 were used as a training data set to build a classification model, and batch 1 was used as the test data set. For the LPS-treated mice, batches 2 and 3 were used to build the classification model, and batch 1 was used as the test data set. In Appendix A, details of the number of mice and the sex used in the study are mentioned. Separate classification models were built for septic mice and mice with inflammation to extract the differences relevant to polymicrobial sepsis and endotoxemia. The PCA-LDA score plot can be seen in Figure 2A for the septic mice and Figure 2B for the LPS-treated mice. The true labels are presented with square (septic) and circle (sham) legends, and the colors of the legend show the classification results: red (septic) and black (sham). The majority of the septic T-lymphocytes (square legend) are predicted as septic (red color) with negative PCA-LDA scores, and the sham cells (square) are predicted as healthy (black color), which have positive PCA-LDA scores with few outliers (Figure 2A). However, the T-lymphocytes from the endotoxemic mice are not as distinctly separated as the septic mice, with no clear separation between the activated and non-activated T-lymphocytes (Figure 2B). The balanced accuracy (Table 2) obtained for the classification model of the T-lymphocytes from the sham and infected mice in the PCI model is 67%.

Although, for the individual experimental batches, a balanced accuracy of ~92% was obtained for the PCI model, when the Raman data from different batches were pooled, the accuracy dropped. The probable cause for the drop in the accuracy could result from the heterogeneity in the degree of inflammation within individual mice and the influence of the experimental variations. However, when a majority vote was considered, i.e., how many Raman spectra of T-lymphocytes were correctly assigned to their respective mice, it gave a balanced accuracy of 100% (Appendix A). Further, the Cohen’s kappa value indicated the PCA-LDA-based separation between the sham and septic mice (0.33) was significant. The significant Cohen’s kappa value implied that the model correctly assigned T-lymphocytes to the sick mice from where they originated and T-lymphocytes from the control mice to their respective hosts. The prediction of independent experimental batch 1 with the sham and infected mice used as prediction data to validate the PCA-LDA classification model is presented in Figure 2E. The confusion matrix of the PCA-LDA prediction model for the test data set is shown in Appendix A. The model correctly predicted the sham and infected mice from the PCI model with an accuracy of 79% (Cohen’s kappa value was 0.57). For the LPS model, a balanced accuracy of 51% (Cohen’s kappa value 0.02) was obtained using the training data set. This low accuracy of the model was also reflected in the prediction of the test data set (balanced accuracy of 40% and Cohen’s kappa value −0.21), where the model was not able to separate between the sham and the endotoxemia mice (Appendix A). The PCA-LDA score plot for the prediction data set of the endotoxemia model is shown in Figure 2F, and the majority of the inflamed cells (square legend) were predicted as healthy (black color).

The non/low distinction of LPS-treated and sham mice agrees with our previous observation [22] where the T-lymphocytes were slightly activated after 24 h post-LPS treatment, yielding a low differentiation accuracy with the Raman spectral model. The severity of infection in the septic mice is reflected in the higher balanced accuracy obtained compared to the LPS mice. The underlying biochemical differences can be visualized using PCA-LDA loading coefficients.

The PCA-LDA loadings are plotted in Figure 2C,D for the PCI and endotoxemia models. In correlation with the PCA-LDA score plot shown in Figure 2A, the positive peaks in the PCA-LDA loadings plot (Figure 2C) are the contributions arising from the T-lymphocytes of the non-infected mice, whereas the negative peaks (Figure 2C) are dominant in the Raman spectra of the T-lymphocytes of the infected mice. The Raman spectral features of the infected mice in the PCI model (Figure 2C) contain Raman peaks arising from the amino acids (554 cm^−1^); nucleic acids (755 cm^−1^, 788 cm^−1^, and 1589 cm^−1^); proteins (1004 cm^−1^, 1346 cm^−1^, 1454 cm^−1^, 1676 cm^−1^, and 2954 cm^−1^); and lipid (1130 cm^−1^, 1376 cm^−1^, 2849 cm^−1^, and 2885 cm^−1^) vibrations [42]. In comparison, in the T-lymphocytes of the sham mice, the presence of nucleic acid vibrations (1190 cm^−1^ and 1331 cm^−1^) is not prominent. However, in the sham mice, the cells display strong Raman peaks at 596 cm^−1^, 659 cm^−1^, and 695 cm^−1^, usually arising from small proteins or amino acids such as cysteine. Amino acids are crucial in maintaining a physiological balance and controlling T-lymphocyte activation [43]. The loss of prominent amino acid vibrations in septic animals probably results from the rearrangement of proteins in activated T-lymphocytes. Although these protein clusters are on the nanoscale, a collective change is captured by Raman spectroscopy [22,43,44]. The different protein vibrations also reflect the changes in the T-lymphocytes’ amino acid compilation between the septic and sham mice. For example, the positive Raman peaks of PCA-LDA loading (Figure 2C) belonging to T-lymphocytes from the sham mice display a different set of protein vibrations (1250 cm^−1^ (amide III beta-sheet); 1265 cm^−1^ (amide III unordered); 1289 cm^−1^, 1430 cm^−1^, and 1628 cm^−1^ (amide I beta-sheet); and 2942 cm^−1^). Hence, the different Raman peaks observed at the low wavenumber region assignable to amino acids and the changed protein vibrations indicate an altered transcriptomic activity. Su et al. and others have shown serum amino acid changes in profiles in patients with sepsis and septic shock [45,46,47]. Similarly, a higher amount of lipids is associated with the crucial support of physiological functions in healthy cells [48,49]. In the current study, the septic T-lymphocytes exhibit the presence of long-chain unsaturated lipids (1130 cm^−1^ and 3011 cm^−1^) compared to the T-lymphocytes from the sham mice (1766 cm^−1^ and 3023 cm^−1^) probably due to the altered lipid metabolism [50,51].

In the endotoxemic mice model, the T-lymphocytes (Figure 2D) displayed a rich content of Raman vibrations. However, a strict distinction between the observed Raman vibrations in the healthy and sick endotoxic mice is hard to discern given that the PCA-LDA model could not distinguish between the endotoxemic and sham mice.

When we look at the overall difference between the septic and endotoxemic mice shown in Appendix A, the majority of the biochemical differences in the T-lymphocytes are present in the fingerprint regions of the amide and nucleic acids, with the contributions coming from the overall changes in the protein and lipid vibrations. It is essential to note that precisely identifying the specific proteins or lipids within the Raman spectra obtained from complex molecules, such as a cell, is not feasible. However, the broad changes observed for the proteins, lipids, and nucleic acids imply that the T-lymphocytes’ metabolic and transcriptomic changes upon exposure to the pathogenic microbes differ from the inflammatory state induced by LPS exposure. There is also a noticeable difference in the healthy mice from both the PCI and LPS models that relates to the heterogeneity among the individual mice and can be attributed to the differences in the mice strains used for the PCI and LPS models (Appendix A). However, the FVB/N and C57BL/6 mice are chemically similar, exhibiting a similar trend of clinical severity scores, as shown in Table 3, when sepsis is induced.

## 3. Material and Methods

### 3.1. Animal Ethic Statement

FVB/N and C57BL/6J mice were bred and housed under specific pathogen-free conditions at the Jena University Hospital’s animal facility. Mice were allowed to adapt to the experimental room’s laboratory conditions. Mice were housed with a 12 h day/night cycle and 1 h dusk/dawn phase. The ambient temperature was maintained at 23 °C and 60–70% humidity. Mice were fed a standard diet. All the experiments done at Jena University Hospital were performed in line with the German legislation on the protection of animals with authorization from the regional animal welfare committee of Thuringia (Thüringer Landesverwaltungsamt), registration numbers 02-046/11 and UKJ-19-010 [22,29].

### 3.2. Peritoneal Contamination Infection (PCI) Mouse Model

Details of the mice experiments have been described in previous publications [52]. The experimental sepsis model of peritoneal contamination and infection (PCI) established complied with the MQTiPSS. Male and female mice between 8 and 12 weeks of age were employed [31]. Human feces, characterized previously regarding their bacterial composition [53], was administered intraperitoneally to FVB/N mice, resulting in polymicrobial sepsis. The control group (sham animals) received 0.9% saline water. Mice were treated with antibiotics (Meropenem, 25 mg/kg body weight (BW)) and ringer acetate (10 mL/kg BW) subcutaneously twice daily starting 6 h after infection. In addition, Metapyrin (2.5 mg/animal) was administered from the beginning onwards every 6 h orally for pain relief. The mice were sacrificed 24 h post-infection. Ten infected (PCI group) and eight uninfected (sham group) mice were investigated.

### 3.3. LPS-Induced Inflammation Mouse Model

The previous publications have described the mice experiment details and the clinical scores [22,29]. Briefly, for the induction of endotoxemia, C57BL/6 mice were injected with LPS or 0.9% saline water (for control mice). Mice were sacrificed 24 h after insult. In three experimental replicates, in each replicate, pooled T-lymphocytes isolated from the spleens of 9–12 mice were investigated.

### 3.4. Splenocyte Isolation

A schematic diagram of the sample preparation is shown in Figure 3. The animals were sacrificed 24 h post-infection, and the spleen organ was harvested. The spleens were placed on a 70 mm cell strainer and gently squashed using a plunger. The cells were then washed with PBS (Biochrome, Berlin, Germany) that was supplemented with 0.5% BSA (PAA laboratories, Coelbe, Germany) and 2 mM EDTA (AppliChem, Darmstadt, Germany). Next, erythrocyte lysis buffer (10 mmol/L KHCO_3_; 150 mmol/L NH_4_Cl; 0.1 mmol/L EDTA) was used to remove the red blood cell contamination of the sample, and the cells were washed again. The T-cell subsets were purified using automated negative magnetic sorting (autoMACS Pro system; Miltenyi Biotech, Bergisch Gladbach, Germany) using standard isolation kit protocols (Miltenyi Biotech, Bergisch Gladbach, Germany), as previously described [29]. For the Raman measurements, the separated splenocytes were fixed with 4% formaldehyde for 10 min at room temperature. The cells were then washed with 1 mL PBS by centrifugation (600 RCF/2000 rpm for 5 min), suspended in 500 uL PBS, and stored at 4 °C until measured. All the cells were measured within 6 h after preparation.

### 3.5. Raman Spectroscopy Investigation of the Splenocytes

The Raman spectra of the splenocytes were recorded using a commercial Raman spectrometer (WITec Alpha 300S, Kroppach, Germany) equipped with a 600 g/mm grating and laser excitation wavelength of 532 nm. The scattered light from the sample was guided through a one hundred-micrometer fiber to the detector. The T-lymphocytes isolated from the PCI mice were placed on the CaF_2_ slide, whereas the T-lymphocytes from the endotoxemic mice were embedded in 100 uL of 0.75% alginate (Sigma-Aldrich, Taufkirchen, Germany) solution on a CaF_2_ slide to allow for measurements in suspension. The cells were immobilized by alginate polymerization using a 0.5 M calcium chloride solution (Sigma-Aldrich) and a washing step with PBS. In both cases, the slides were immersed in a petri dish filled with sterile, filtered water. The differences in the sample measurement conditions did not influence the results, as separate Raman models were built for the septic and endotoxemic mice. A comparative analysis was done to highlight the changes in the T-lymphocytes from the septic mice and endotoxemic mice. The samples were measured using a 60x/NA 1.0 (laser spot size ~0.3 µm) water immersion objective (Nikon). Raman spectra of the PCI T-lymphocytes were screened by recording single spectra/cell with a 0.5 s integration time for each spectrum and a laser power of ~65 mW before the objective (~30 mW on the sample). About 625–1025 cells were recorded for each sample. The Raman spectra of the LPS T-lymphocytes were recorded in imaging mode with a step size of 0.3 µm, 1 s integration time/spectrum, a laser power of ~36 mW before the objective (~30 mW on the sample), and laser spot size of ~0.3 µm. About 300–400 spectra/cell were recorded, and 10–20 cells were measured. Multiple experimental replicates were performed.

### 3.6. Raman Spectral Data Analysis

The data preprocessing and statistical analysis were done using RAMANMETRIX Leibniz-IPHT 2020 software version 0.4.3 [54]. Preprocessing of the Raman spectra included removing cosmic spikes using the k-means cluster algorithm followed by spectral background subtraction using the sensitive nonlinear iterative peak (SNIP) and vector normalization. The Raman spectra were clipped (excluding the spectral regions 1800–2800 cm^−1^ and above 3050 cm^−1^), and the spectral range of 400–3050 cm^−1^ was used for further analysis. The preprocessed spectra were further processed by calibrating the wavenumber axis using the Raman spectra of 4-acetamidophenol, used as the standard reference. The standard reference substance was measured each day before recording the Raman spectra of the cells. The mean Raman spectra were generated. The graphical plots of the data were reconstructed using Origin 2020 software.

### 3.7. Raman Spectral Model for PCI Mice

The classification models were built firstly for individual batches 1 to 4 (batch 5 consisted only of sick animals; thus, no model was built) to gain an overview of the batch-to-batch heterogeneity. Three different classification methods were used to build the Raman spectral-based models—namely, a principal component analysis followed by a linear discriminant analysis (PCA-LDA), a principal component analysis followed by random forest (PCA-RF), and a principal component analysis followed by support vector machine (PCA-SVM)—and 10-fold cross-validation was performed to calculate the mean sensitivity, termed the balanced accuracy. In the second step, the PCA-LDA method was used for the combined analysis of all the acquired data. Then, the data were divided into a training data set and a test data set. The training data set included preprocessed Raman spectra collected from batches 2 to 5 with 12,219 spectra, aggregated by 10 spectra from each type. The number of spectra for aggregation was optimized for the optimum separation of sick and healthy animals. First, the PC analyses were performed, and 5 PCs were used as input for the LD analysis. Then, the training model was internally cross-validated using the batchwise cross-validation method, and a balanced accuracy, sensitivity, and specificity were obtained for the PCA-LDA model. Finally, the training model was externally cross-validated using the test Raman spectral data from batch 1 (3549 spectra).

### 3.8. Raman Spectral Model for LPS Mice

The classification models were also built firstly for individual batches 1 to 3 to gain an overview of the batch-to-batch heterogeneity (Appendix A). The models were built using the same classification methods as in the PCI model: PCA-LDA, PCA-RF, and PCA-SVM. Again, 10-fold cross-validation was performed, and the balanced accuracy for each model was calculated.

In the second step, the PCA-LDA method was used for the combined analysis of all the acquired data. The data were divided into a training data set and a test data set. The training data set included preprocessed Raman spectra collected from batches 1 and 3 with 36,177 spectra, aggregated by 12 spectra, and an optimum number of spectra giving the maximum separation between the sick and healthy animals. First, the PC analyses were performed, and 7 PCs were used as input for the LD analysis. Next, the training model was internally cross-validated using the batchwise cross-validation method, and the balanced accuracy, sensitivity, and specificity of the PCA-LDA model were determined. Then, the training model was externally cross-validated using the test data set consisting of the Raman spectral data from batch 2 (16,797 spectra).

## 4. Conclusions

Raman spectroscopy was used to analyze the splenocytes of mice with systemic inflammation induced by lipopolysaccharide or bacterial sepsis within 24 h. Our findings disclosed significant changes in the chemical composition of the splenocytes, primarily in the DNA region, and minor changes in the amino acids and lipoproteins were also observed, as expected during sepsis. While the extent of these changes varied between individual mice, the overall qualitative changes were highly reproducible. The Raman models showed good discrimination of the septic mice against the control, but poor discrimination was obtained between endotoxemic mice and their respective controls. The accuracy of the classification model for the combined batches for PCI was 67.4% and that of LPS was 54.6%. However, the classification accuracy for individual batches for the PCI model ranged from 89% to 96%, while, for the LPS model, it ranged from 60 to 90%. This study gives an idea about the Raman spectra profile and the changes occurring within 24 h in animals with systemic inflammation and allows an understanding of the disease-relevant changes, which can be translated into further studies within humans.

## Figures and Tables

**Figure 1 ijms-24-12027-f001:**
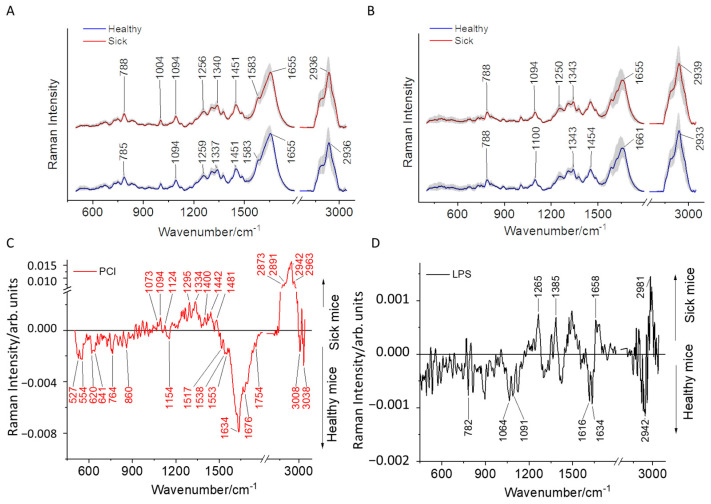
Mean Raman spectra, along with the standard deviation of T-lymphocytes isolated from the spleens of (**A**) mice with sepsis due to peritoneal contamination infection (red) and the sham (blue) and (**B**) endotoxemic mice (red) and the sham (blue) mice. Raman difference spectra are generated by subtracting. (**C**,**D**) Raman difference spectra generated by subtracting the Raman spectra of T-lymphocytes isolated from the spleens of sham mice from (**C**) mice with sepsis due to peritoneal contamination infection and (**D**) endotoxemic mice.

**Figure 2 ijms-24-12027-f002:**
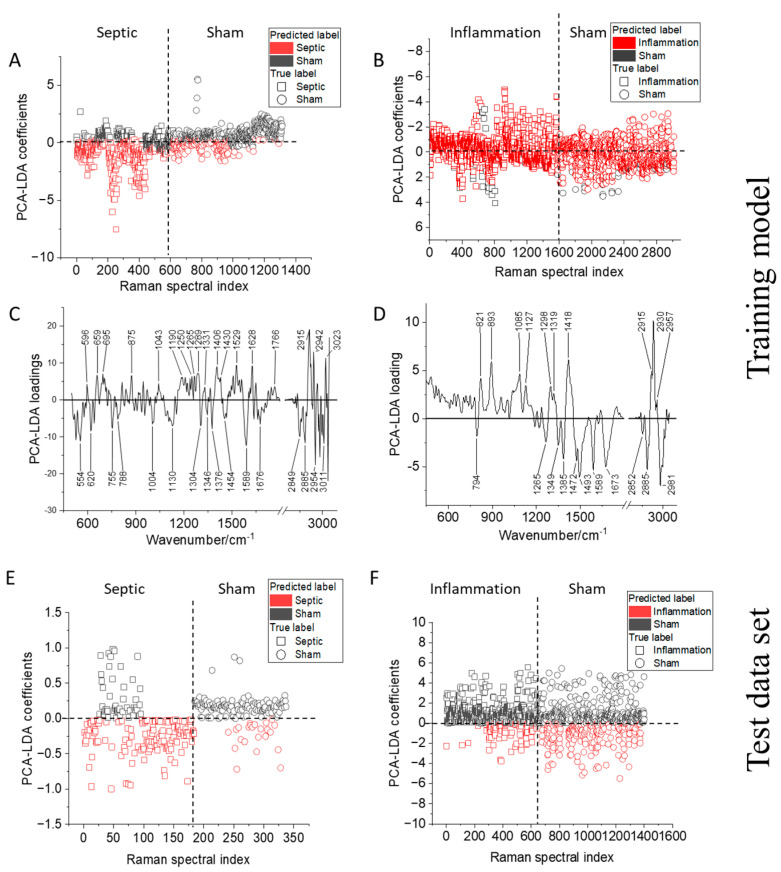
Optimized Raman model to differentiate T-lymphocytes isolated from the spleens of (**A**) septic (PCI mice) 24 h post-infection and (**B**) mice with inflammation (endotoxemia) 24 h post-insult, with PCA-LDA loading 1 obtained for (**C**) the PCI and (**D**) LPS models. Cross-validation using the test data set is shown for (**E**) the PCI and (**F**) LPS models. The index in (**A**,**B**,**E**,**F**) refers to the number of Raman spectra used for building the classification model. The true labels are presented with square (septic) and circle (sham) legends, and the colors of the legends show the classification/prediction results: red (septic) and black (sham).

**Figure 3 ijms-24-12027-f003:**
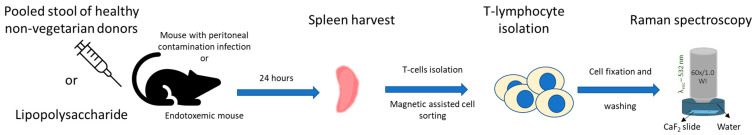
Experimental flow chart of T-cell isolation and analysis from the murine spleens.

**Table 2 ijms-24-12027-t002:** Comparison of the balanced accuracy for differentiating T-lymphocytes from healthy and septic/inflammation mice.

Mice Models	Balanced Accuracy */Cohen’s Kappa			
	PCA-LDA Model for Classification Using the Training Data Set		PCA-LDA Model for Prediction Using the Test Data Set	
	Individual Spectra	Majority Vote	Individual Spectra	Majority Vote
**PCI**	0.67/0.33	1/1	0.79/0.57	1/1
**LPS**	0.51/0.02	0.5/0	0.40/−0.21	0.5/0

* Balanced accuracy is the mean sensitivity.

**Table 3 ijms-24-12027-t003:** Comparison of the clinical features of the FVB/N and C57BL/6 septic mice.

Clinical Feature 24 h Post-Infection	FVB/N Septic Mice	C57BL/6 Septic Mice (Previously Published Data in Ref. [29])
Clinical severity score	2.2	2.0
Mortality 10 days post-infection (%)	78%	56%

## Data Availability

The data will be made available upon request.

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
