# Peer review of "Raman Spectroscopy Profiling of Splenic T-Cells in Sepsis and Endotoxemia in Mice"

_ijms, 2023, doi:10.3390/ijms241512027_

Round 1

Reviewer 1 Report

The manuscript “Raman spectroscopy profiling of splenic T-cells in sepsis and endotoxemia in mice”, by I.E. Osadare et al., investigates Raman spectra measured for two different mice types (FVB/N and C57BL/6) for which sepsis and endotoxemia were induced by means of administration of human feces and treatment with LPS, respectively. The authors carried out Raman measurements for cell samples from the spleen of endotoxemic and septic mice. In particular, they found biochemical differences mainly in the Raman spectra of septic samples with respect to control ones and they used classification statistical techniques for discriminating the sick samples from control ones.

Although my experience is related to Raman spectroscopy and I am not an expert in mouse models, the manuscript presents some drawbacks in the investigation methodology and also several aspects that need to be improved before being re-submitted for publication. I then provide descriptions of the major drawbacks in the manuscript and some indications that could be useful in drafting a new manuscript.

 Major revisions

·    The authors claimed that “The aim was to gain insight into the potential of Raman spectroscopy for distinguishing biochemical alterations in endotoxemia or sepsis”. It should be clearly stated in the Conclusions section that discrimination of sepsis versus control was obtained, but discrimination of endotoxemia versus control was poor (balanced accuracy 40% for test set). If, on the other hand, the aim is to discriminate cells affected by sepsis from those affected by endotoxemia, this is not possible because two different mouse models have been investigated.

·   In this regard, it is necessary to justify why two different mouse models (FVB/N and C57BL/6) were used to investigate the two pathologies. It would be much better to choose only one model, because the Raman spectra of the two types of control are different from each other, as evident in Figure S2 (black line).

·      Figures S1 and S2 should be moved from the Supplementary Information to the main article, because it is explanatory for the data analysis to show the spectral difference between the two cell murine models, especially as for the healthy spectra is concerned.

·       Figure 2: the spectral differences between the LPS-treated (mean) spectrum and the corresponding control (mean) spectrum do not seem very significant in any spectral range, at least when compared to the spectral differences obtained with cells infected by sepsis. Are the authors certain that the spectral differences for LPS spectra (black line in Figure 2) are not due to noise? (as an example, in the region of CH stretching the contribution of noise seems to be important).

·     Figure 3: this figure does not provide useful information, as it is the difference between a Raman spectrum from endotoxin mice and a Raman spectrum from septic mice, but the two spectra that are subtracted from each other refer to mouse models whose controls are characterized by different Raman spectra. As an example, the authors declare at lines 272-273 that “The T-lymphocytes from endotoxemic mice exhibit rich CH stretching regions compared to the septic mice”: this occurs because the control spectrum of C57BL/6 has larger CH stretching signals than the control spectrum of FVB/N (see Figures S1 and S2). Therefore, the Figure 3 should be removed and the comments of the obtained results (lines 263-288 of the present manuscript) should be modified and adapted to describe Figure 2.

·      Lines 349-375: spectral positions are mentioned which refer to a Figure not present in the manuscript (“Figure 4A below”). So, it takes a lot of effort and follow the description. Perhaps it is necessary to insert the figure that is described?

·     Lines 380-387: several mentioned spectral positions are not labelled in Figure 4D: therefore, it is not easy to follow the description of this figure. Moreover, as the authors declare, there is no evident distinction between controls and sick in the score plot in Figure 4B: therefore, the description of the loading plot is not very useful.

 Minor revisions

·       Section 2.4: it would be better to report the size of the laser spot focused on the cell sample. This is important to understand if a single Raman measurement involves several cells, or a single cell, or subcellular components (nucleus, cytoplasm, ...).

·       Lines 234-235: was the subtraction done for the mean spectra? This should also be specified in the caption of Figure 2.

·       Line 244: the spectral signals at 764, 1295 and 1553 cm-1 are attribute to proteins in Figure 2 and Table 1.

·       Line 249: please replace “…amide I, the CH…” with “…amide I and CH…”.

·       Line 294: PCA is an unsupervised statistical method, whereas PCA-LDA, PCA-RF and PCA-SVM are supervise statistical methods.

·       Figure 4A and 4B and lines 311-312: T-lymphocytes from septic mice have negative scores in the Figure, contrary to what reported in the text. Similarly, the sham cells have positive scores.

·       Line 313: replace “blue” with “black”.

·   Legend of Figure 4A and 4B: check the symbols used for true label septic and inflammation: it is probably necessary to insert a hollow red circle instead of the black rectangle.

·    Figure 4A and 4B: specify what the "index" title on the horizontal axis refers to. Explain, in the caption, which measures are related to the training set and which are related to the test set.

·    Figure 4C and 4D: please specify which loading is reported in the vertical axis (loading 1?).

At several points in the manuscript it is necessary to rewrite the sentences in a more appropriate way.

Reviewer 2 Report

Please find an attached file.

Round 2

Reviewer 1 Report

The manuscript improved after the revisions by the authors. However, a couple of points still need to be clarified. In particular, the answers given by the authors to point 5 and point 2 are not satisfactory.

Point 5. Both in the text (lines 295-297) and in the Figure 3 caption it is reported that Figure 3C represents the Raman difference spectrum generated by subtracting mean spectrum of T‑lymphocytes isolated from the spleen of endotoxin mice from septic mice. As I've commented in my previous revision, this difference doesn't provide useful information. Indeed, the larger spectral differences shown in Figure 3C (i.e. in the amide I and CH stretching spectral ranges) are already evident in the controls (see Figure 3D) and, consequently, they are not due to the different pathology but to the different mouse models.

Point 2. It was not explained why two different mouse models were used. This choice must be justified, otherwise there is a methodological error in this investigation.

Author Response

Response to Reviewer 1 Comments

Comments and Suggestions for Authors

The manuscript improved after the revisions by the authors. However, a couple of points still need to be clarified. In particular, the answers given by the authors to point 5 and point 2 are not satisfactory.

Point 5: Both in the text (lines 295-297) and in the Figure 3 caption it is reported that Figure 3C represents the Raman difference spectrum generated by subtracting mean spectrum of T‑lymphocytes isolated from the spleen of endotoxin mice from septic mice. As I've commented in my previous revision, this difference doesn't provide useful information. Indeed, the larger spectral differences shown in Figure 3C (i.e. in the amide I and CH stretching spectral ranges) are already evident in the controls (see Figure 3D) and, consequently, they are not due to the different pathology but to the different mouse models.

Response 5: We have merged sections 3.1 and 3.2. The mean Raman spectrum from Figure 3 has been moved to Figure 2 and the difference spectrum from Figure 3 has been moved to supplementary.

In our opinion, despite the larger spectral differences evident also in the controls, the differences observed are not only limited to the different mouse strains. The differences observed in the amide I and CH stretching spectral ranges for the difference Raman spectrum generated between the control mice from PCI and LPS models are also observed in some of the control mice within the given strain. For example, in new supplementary Figure S2C the difference between healthy mice in batch 1 - batch 3 exhibit large changes in the amide I and CH stretching spectral ranges. Similarly, is seen for control mice from other batches and also in the control from the LPS model.

Point 2: It was not explained why two different mouse models were used. This choice must be justified, otherwise there is a methodological error in this investigation.

Response 5: Thank you for raising your concern about the use of two different mouse models in our investigation. Allow us to elaborate on the rationale behind this decision in order to address your query.

The selection of mouse models in biomedical research is a crucial aspect, as it can significantly impact the validity and reliability of the findings. In our study, we employed two different mouse models, namely the C57BL/6J (B6) and FVB/N mice, due to specific characteristics exhibited by each strain.

Firstly, the FVB/N mice were chosen for their unique qualities in relation to the peritoneal contamination infection model. These mice display a reduced sensitivity towards this particular type of infection and a reduced “innate response” compared to B6. Since FVB/N mice rely more on the adaptive immune response, that is investigated here, than B6 in this particular model their host-response kinetics mimics the human situation more.

Conversely, the host response of B6 mice to LPS is considered most similar to the human response (1–8).

Our aim was to mimic the infection and inflammation disease and explore the chemical changes during acute stage using Raman spectroscopy.

By including the FVB/B6 mice in our experimental design, we aimed to capture and compare the biochemical differences during the acute infection/inflammation phase induced by the infectious/toxic stimuli respectively. Both models are compared to their respective baseline (sham treated animal). This comparative analysis provided us with valuable insights into the specific mechanisms and responses associated with LPS-induced endotoxemia versus peritoneal infection.

In conclusion, the decision to utilize both the B6 and FVB/N mouse models in our investigation was based on their unique characteristics and the specific objectives of our study. We acknowledge that this choice may seem unconventional at first glance, but it was carefully justified considering the limitations and advantages associated with each strain. We believe that this approach contributes to a more comprehensive understanding of the biochemical differences captured by Raman spectroscopy during the acute infection/inflammation phase in both mouse strains.

We have included a few reference supporting our above justification with respect to FVB and B6 mice. If you have any further questions or require additional clarification, please feel free to ask.

References

  1. Hoshino, K.; Takeuchi, O.; Kawai, T.; Sanjo, H.; Ogawa, T.; Takeda, Y.; Takeda, K.; Akira, S. Cutting edge: Toll-like receptor 4 (TLR4)-deficient mice are hyporesponsive to lipopolysaccharide: evidence for TLR4 as the Lps gene product. J Immunol 1999, 162 (7), 3749–3752.
  2. Skelly, D. T.; Hennessy, E.; Dansereau, M.-A.; Cunningham, C. A systematic analysis of the peripheral and CNS effects of systemic LPS, IL-1β, corrected TNF-α and IL-6 challenges in C57BL/6 mice. PLoS ONE 2013, 8 (7), e69123.
  3. Shi, H.; Kokoeva, M. V.; Inouye, K.; Tzameli, I.; Yin, H.; Flier, J. S. TLR4 links innate immunity and fatty acid-induced insulin resistance. The Journal of clinical investigation 2006, 116 (11), 3015–3025.
  4. Pullen, L. C.; Park, S. H.; Miller, S. D.; Dal Canto, M. C.; Kim, B. S. Treatment with bacterial LPS renders genetically resistant C57BL/6 mice susceptible to Theiler's virus-induced demyelinating disease. J Immunol 1995, 155 (9), 4497–4503.
  5. Kishimoto, S.; Takahama, T.; Mizumachi, H. In vitro immune response to the 2,4,6-trinitrophenyl determinant in aged C57BL/6J mice:changes in the humoral immune response to, avidity for the TNP determinant and responsiveness to LPS effect with aging. J Immunol 1976, 116 (2), 294–300.
  6. Poltorak, A.; Merlin, T.; Nielsen, P. J.; Sandra, O.; Smirnova, I.; Schupp, I.; Boehm, T.; Galanos, C.; Freudenberg, M. A. A point mutation in the IL-12R beta 2 gene underlies the IL-12 unresponsiveness of Lps-defective C57BL/10ScCr mice. J Immunol 2001, 167 (4), 2106–2111.

Round 3

Reviewer 1 Report

I am satisfied with the authors' response to point 2. Consequently, the considerations that the authors included in their response to my comment should be summarized in the main text of the manuscript, in order to justify (to the readers) the use of two different mouse models.

Instead, I am not satisfied with the authors' response to point 5. The spectra shown in Figure S1A and S1B are almost overlapping. So, in my opinion, the differences between PCI and LPS are already present in the control samples. Furthermore, the authors also declare that spectral differences similar to those shown in Figure S1A in the amide I and CH stretching ranges are already present in the difference spectra between control cells of various experimental batches. Therefore, such difference is due to the intracellular heterogeneity of the different cell components, as also declared by the authors. Therefore, also the commentary text of Figure S1 in the main manuscript is not relevant.

Minor revision

Caption of Figure 2: remove “Raman … subtracting” at line 248.